# Using the Water and Sanitation for Health Facility Improvement Tool (WASH FIT) in Zimbabwe: A Cross-Sectional Study of Water, Sanitation and Hygiene Services in 50 COVID-19 Isolation Facilities

**DOI:** 10.3390/ijerph18115641

**Published:** 2021-05-25

**Authors:** Mitsuaki Hirai, Victor Nyamandi, Charles Siachema, Nesbert Shirihuru, Lovemore Dhoba, Alison Baggen, Trevor Kanyowa, John Mwenda, Lilian Dodzo, Portia Manangazira, Musiwarwo Chirume, Marc Overmars, Yuhei Honda, Ajay Chouhan, Boniface Nzara, Placidia Vavirai, Zvanaka Sithole, Paul Ngwakum, Shelly Chitsungo, Aidan A. Cronin

**Affiliations:** 1United Nations Children’s Fund (UNICEF), Harare, Zimbabwe; movermars@unicef.org (M.O.); yhonda@unicef.org (Y.H.); ajaychouh1@gmail.com (A.C.); bnzara@unicef.org (B.N.); pvavirai@gmail.com (P.V.); pngwakum@unicef.org (P.N.); schitsungo@unicef.org (S.C.); 2Ministry of Health and Child Care, Government of Zimbabwe, Harare, Zimbabwe; victornyamandi@gmail.com (V.N.); csiachema@gmail.com (C.S.); jonmwenda@gmail.com (J.M.); dodzolilian@yahoo.com (L.D.); directoredc@gmail.com (P.M.); chirumemusiwarwo@gmail.com (M.C.); 3Ministry of Lands, Agriculture, Fisheries, Water, Climate and Rural Settlement, Government of Zimbabwe, Harare, Zimbabwe; nshirihuru@gmail.com (N.S.); dhobal@africau.edu (L.D.); 4Action Contre la Faim, Harare, Zimbabwe; washco@zw-actionagainsthunger.org; 5World Health Organization, Harare, Zimbabwe; kanyowat@who.int (T.K.); sitholez@who.int (Z.S.); 6United Nations Children’s Fund (UNICEF), New York, NY 10017, USA; acronin@unicef.org

**Keywords:** water, sanitation, hygiene, health care waste management, infection prevention and control, WASH, health care facility

## Abstract

The availability of water, sanitation and hygiene (WASH) services is a key prerequisite for quality care and infection prevention and control in health care facilities (HCFs). In 2020, the COVID-19 pandemic highlighted the importance and urgency of enhancing WASH coverage to reduce the risk of COVID-19 transmission and other healthcare-associated infections. As a part of COVID-19 preparedness and response interventions, the Government of Zimbabwe, the United Nations Children’s Fund (UNICEF), and civil society organizations conducted WASH assessments in 50 HCFs designated as COVID-19 isolation facilities. Assessments were based on the Water and Sanitation for Health Facility Improvement Tool (WASH FIT), a multi-step framework to inform the continuous monitoring and improvement of WASH services. The WASH FIT assessments revealed that one in four HCFs did not have adequate services across the domains of water, sanitation, health care waste, hand hygiene, facility environment, cleanliness and disinfection, and management. The sanitation domain had the largest proportion of health care facilities with poor service coverage (42%). Some of the recommendations from this assessment include the provision of sufficient water for all users, Menstrual Hygiene Management (MHM)- and disability-friendly sanitation facilities, handwashing facilities, waste collection services, energy for incineration or waste treatment facilities, cleaning supplies, and financial resources for HCFs. WASH FIT may be a useful tool to inform WASH interventions during the COVID-19 pandemic and beyond.

## 1. Introduction

The availability of water, sanitation and hygiene (WASH) services is a key prerequisite for quality care and infection prevention and control (IPC) in health care facilities (HCFs) [1,2]. In low- and middle-income countries, however, WASH service coverage in HCFs remains low [3]. A study of 78 countries estimated that 50% of HCFs do not have piped water connection, 33% do not have improved sanitation facilities, and 39% do not have handwashing soap [3]. The World Health Organization (WHO) and the United Nations Children’s Fund (UNICEF) Joint Monitoring Programme for Water, Sanitation and Hygiene (JMP) also reported that in 2019, 1.8 billion people did not have access to basic water services in HCFs, and 800 million people relied on HCFs without improved sanitation facilities [4]. In Zimbabwe, basic water services were available in 81% of HCFs, but only 17% and 58% of HCFs had basic sanitation and hygiene services, respectively [4].

Previous studies suggested that limited WASH services and hygiene practices in HCFs have negative implications on patient satisfaction [5], care-seeking behaviors [5], healthcare-associated infections (HCAIs) [6,7,8,9,10,11], and patient dignity [12]. While WASH is not the only risk factor, the estimated pooled prevalence of HCAIs ranges from 7% in high-income countries to 15.5% in low-income countries [9,10,11]. Inadequate access to water, sanitation, and a clean environment is also indirectly linked to antimicrobial resistance (AMR) by increasing the risk of HCAIs and the demand for antibiotics [13].

In May 2019, the World Health Assembly passed a resolution to accelerate global efforts on WASH in HCFs [14]. This resolution led to a subsequent global meeting where countries presented their national commitments with concrete actions [15]. Zimbabwe made key country commitments, including the development of a national WASH in HCFs plan, standards, and targets [16]. In January 2020, the Ministry of Health and Child Care officially approved the national WASH in HCF Taskforce with representation from various departments of the Ministry of Health and Child Care, the Ministry of Local Government and Public Works, the Ministry of Lands, Agriculture, Water and Rural Resettlement, UN agencies, civil society organizations (CSOs), and the Health Professions Authority (Figure 1).

In 2020, the COVID-19 pandemic highlighted the importance and urgency of enhancing WASH coverage in HCFs [17]. Globally, over 166.7 million cases and 3.4 million deaths were reported as of 23 May 2021 [18]. Zimbabwe has recorded a total of 38,679 cases and 1586 deaths since the first confirmed case in mid-March 2020 [19]. To minimize the risk of transmission at the designated HCFs for COVID-19 patients, the Government of Zimbabwe, with support from UNICEF and CSOs, has implemented a package of WASH interventions in 50 HCFs. This WASH response package included WASH assessment, IPC training for health care staff, provision of IPC and WASH supplies (e.g., soap, handwashing stations, cleaning materials), and repair of WASH facilities.

To date, few studies have elucidated if the existing WASH assessment methodology can be useful under the COVID-19 response and how WASH in HCF interventions can be implemented by health and WASH sectors without a siloed approach [20]. Using the Water and Sanitation for Health Facility Improvement Tool (WASH FIT) as the guiding assessment methodology [21], this study aimed to describe the status of WASH services in 50 COVID-19 isolation facilities and discuss how WASH in HCF Taskforce members may contribute to addressing key gaps identified.

## 2. Materials and Methods

### 2.1. WASH FIT Assessment

WASH FIT was developed as a multi-step framework to inform the continuous monitoring and improvement of WASH services, identify gaps in policies and standards, foster an enabling environment by sharing responsibilities among key stakeholders, enhance daily management and operation of HCFs, and engage community members to represent the voice of service recipients [21]. The main intended users of WASH FIT include HCF managers and staff (e.g., doctors, nurses, non-medical officers), and additional personnel from local government, communities, and CSOs. Based on the assessment and monitoring, WASH FIT users may develop WASH improvement plans and implement specific interventions to address any gaps identified.

WASH FIT covers six domains and sub-domains: water, sanitation, health care waste, hand hygiene, facility environment, cleanliness and disinfection, and management. For each domain, a list of pre-defined indicators were available. While WASH FIT was developed mainly for primary health care facilities, it could be adapted for larger HCFs [21].

### 2.2. Data Collection

The Government of Zimbabwe, UNICEF, and CSOs conducted WASH FIT assessment in 50 HCFs to collect key information on the status of WASH services. The number of target HCFs was set as 50 based on the available financial and human resources. To select 50 HCFs, the taskforce team had consultations with the national and sub-national government counterparts to identify priority health care facilities designated as COVID-19 isolation facilities, and reviewed the availability and operational presence of implementing partners in respective provinces. Data were collected through direct observations and consultations with HCF staff from June to September 2020 by three assessment teams consisting of local government representatives, health facility staff, and CSO staff.

Each assessment team recorded scores with a checklist for a total of 66 indicators. Of these, 29 are categorized as essential indicators, and 37 are advanced indicators [21]. Potential scores for each indicator range from 0 to 2. Thus, the maximum possible score for the assessment is 132 points (i.e., 2 points × 66 indicators). Details on the scoring criteria and assessment forms for each domain and indicator can be found elsewhere [21]. The scores were entered into a Microsoft Excel file, and UNICEF consolidated data files from the assessment teams. Additional background information on HCFs, including the number of beds, types of water sources, and type of sanitation facilities, was also collected.

### 2.3. Data Analysis

A consolidated raw data file was cleaned and analyzed with STATA 14 [22]. Descriptive analyses were conducted to characterize the IPC and WASH status in 50 HCFs across the WASH FIT domains. To estimate average WASH FIT scores by the size of HCFs, we used the number of beds as a criterion to categorize HCFs as small (<100 beds), medium (100–199 beds), or large (200 beds or more). Based on the mean of WASH FIT scores from 50 HCFs, the status of each indicator result was rated as poor (WASH FIT average score <1.0), fair (1.0–1.5), or good (>1.5).

Additionally, a sum of scores for each domain was calculated and divided by the full score of a given domain to estimate the percentage of achievement. For instance, the water domain includes 15 indicators, and a full score would be 30 points (i.e., 2 points*15 indicators). If a HCF scores 15 points in the water domain, the percentage would be 50% (i.e., 15 divided by 30). Based on the percentage, HCFs were classified into three categories: poor (below 50%), fair (50–75%), and good (over 75%).

## 3. Ethical Review

The main purpose of WASH FIT data collection was to inform COVID-19 preparedness and response interventions, and this study analyzed secondary data without any personally identifiable information or individual data. Based on the ethical requirements for human research in Zimbabwe, formal ethical approval was not required. Nonetheless, this study did not name any of the specific HCFs included in this study to minimize the risks of any unintended consequences.

## 4. Results

Table 1 summarizes the descriptive characteristics of 50 HCFs in this study. The WASH FIT assessment covered seven rural provinces (Manicaland, Mashonaland Central, Mashonaland East, Masvingo, Matabeleland North, Matabeleland South, and Midlands) and one metropolitan area (Bulawayo) in Zimbabwe. Only eight of these HCFs had 200 beds or more, and the other HCFs were either small (n = 22) or medium-sized (n = 20). Most of the HCFs were public hospitals or clinics, but a limited number of private, faith-based HCFs were included. The majority of HCFs had access to public taps or piped connections as the main source of water, and 86% of the HCFs had flush toilets. The average number of daily inpatients and outpatients was 37.0 and 67.9, respectively.

Table 2 presents a list of WASH FIT essential indicators, their average scores for the 50 HCFs by the size of HCF, and the rating of indicator status. The number of essential indicators ranged from two indicators for the hand hygiene domain to seven indicators for the facility environment, cleanliness and disinfection domain. The comprehensive list of WASH FIT indicators and their average scores can be found in Appendix A.

### 4.1. Water Domain

The water domain included four essential indicators on water availability, accessibility, and storage. The availability of water supply piped into HCFs or on premises, the presence of reliable drinking water stations, and the presence of drinking water storage have been rated as fair. The availability of sufficient water for all users has been rated as poor. The average score for 50 HCFs was highest on the availability of improved water supply piped into HCFs or on premises at 1.44. It was lowest on the availability of water services, with a sufficient quantity for all users at 0.94. Compared to small and medium-sized HCFs, large HCFs had a higher score across all water domain indicators.

### 4.2. Sanitation Domain

The sanitation domain included six essential indicators on the availability of usable toilets, the separation of improved latrines by personnel type and sex, the presence of menstrual hygiene management (MHM)-friendly and disability-friendly sanitation facilities, and the presence of handwashing stations within five meters of latrines. The overall ratings indicated three indicators as fair, two as poor, and one as good. The score on MHM-friendly and disability-friendly sanitation facilities was particularly low at 0.56 and 0.34, respectively. Regardless of the size of HCFs, the scores on these indicators were lower than 1. Large HCFs scored lower than smaller HCFs on the presence of sex-separated improved latrines.

### 4.3. Health Care Waste Domain

The health care waste domain included six essential indicators on trained personnel, waste collection, waste segregation, waste disposal, waste treatment, and energy availability. The indicator scores on trained personnel, waste disposal practices, waste segregation at waste generation points, and waste treatment technology have been rated as fair. The other two indicators on waste collection containers and energy availability have been rated as poor. The average score on the presence of waste collection containers at waste generation points in small HCFs was 0.77, while medium-sized and large HCFs scored 1.05 and 1.38, respectively. The average score on the availability of energy for waste treatment technologies was lower than 1.0 across all sizes of HCFs.

### 4.4. Hand Hygiene Domain

The hand hygiene domain included two indicators on the availability of functioning hand hygiene stations at all points of care and the presence of hand hygiene promotion materials at key locations in HCFs. Both indicators were rated as fair. Medium-sized HCFs scored 1.40 for these indicators, higher than small and large HCFs.

### 4.5. Facility Environment, Cleanliness and Disinfection Domain

The facility environment, cleanliness and disinfection domain included seven indicators on environmental cleanliness, general lighting, floor cleanliness, cleaning supplies, staff skills on cleaning and disinfection practices, and insecticide-treated mosquito nets. Four indicators on general lighting, cleanliness of floors and surfaces, cleaning supply availability, and staff skills on cleaning and disinfection were rated as fair. Two indicators on the availability of gloves, overalls, boots, and insecticide-treated mosquito nets were rated as poor. The remaining indicator on environmental cleanliness and external fencing was rated as good. The overall average score on the availability of personal protective equipment (PPE) for cleaning and waste disposal staff was 0.88, and small and large HCFs scored 0.77 and 0.75, respectively.

### 4.6. Management Domain

The management domain included four indicators on the presence of WASH FIT or other quality improvement plans, the availability of adequate annual budget for HCFs, the presence of a diagram of the facility management structure, and staff availability for cleaning and WASH. Two indicators were rated as poor, and the other two indicators were rated as fair. The overall average score was lowest on the availability of annual budget for HCFs at 0.72. For the same indicator, large HCFs scored 1.25 while small and medium-sized HCFs scored 0.50 and 0.75, respectively.

### 4.7. WASH FIT Summary Measures

Figure 2 presents the proportion of HCFs with poor, fair, and good WASH service coverage by the size of HCFs. For the water domain, the overall proportion of HCFs with good WASH service coverage was 18% (n = 9), ranging from 9% in small HCFs to 50% in large HCFs. Across all sizes of HCFs, the sanitation domain had the largest proportion of HCFs, with poor service coverage at an overall average of 42% (n = 21). For the health care waste domain, at least 70% of medium-sized and large HCFs had fair or good service coverage while small HCFs had a proportion of poor service coverage at 32%. For the hand hygiene domain, the proportion of HCFs with poor service coverage was 32%, ranging from 25% in medium-sized HCFs to 50% in large HCFs. For the facility environment, cleanliness and disinfection domain, at least 70% of HCFs, regardless of their size, achieved fair or good service coverage. For the management domain, none of the large HCFs were rated as poor coverage while poor coverage reached 36% and 35% in small and medium-sized HCFs, respectively.

## 5. Discussion

This study described the status of WASH services in 50 HCFs designated as COVID-19 isolation facilities by using the WASH FIT as a standardized assessment methodology. The findings revealed that, on average, over 25% of HCFs did not have adequate WASH services in water, sanitation, health care waste, hand hygiene, facility environment, cleanliness and disinfection, or management domains. Nine out of 29 essential WASH FIT indicator results were rated as poor on WASH service coverage in HCFs. To reduce the risk of COVID-19 and other infectious disease transmission in the isolation facilities, immediate actions are needed to provide sufficient water for all users, MHM-friendly and disability-friendly sanitation facilities, handwashing facilities, waste collection, energy for incineration or waste treatment facilities, cleaning supplies, and financial resources for HCFs to manage WASH facilities.

In Zimbabwe, WASH in HCF Taskforce members will leverage their unique expertise to address key gaps identified by WASH FIT assessment. For example, the Ministry of Local Government and Public Works, the Ministry of Lands, Agriculture, Water and Rural Resettlement, UNICEF, and CSOs may prioritize and facilitate rehabilitation of WASH infrastructure and strong reinforcing of good hygiene practices in close liaison with HCF management teams and various departments of the Ministry of Health and Child Care. The Department of Quality Assurance and WHO may support HCFs on their compliance with WASH standards and guidelines. The National Institute of Health Research may conduct field research to explore new technologies for efficient waste treatment, management, and improved handwashing practices. The Department of Epidemiology and Disease Control and the Department of Environmental Health Services may play a vital role in ensuring that WASH in HCF interventions are linked with other key national priorities, such as cholera elimination by 2028 [23], implementation of the national development strategy [24], and other WASH interventions in communities and schools.

One analytical contribution of this study was to classify HCFs by key cut-off points (i.e., poor = <50%, fair = 50–75%, good = 75% or higher). This analysis contributed to the identification of specific gaps and priority intervention areas across WASH FIT domains and sub-domains. The cut-off points remain flexible and may be modified depending on the aim and purpose of WASH FIT users. While WASH FIT may not be a comprehensive tool [25], the potential utility and application of WASH FIT have been well-documented [20,26,27,28]. Moreover, WASH FIT will be further updated to accommodate additional topics, such as climate, occupational health, and gender [29]. Thus, WASH FIT may be scaled up in Zimbabwe and other countries as a key component of a WASH improvement methodology in HCFs.

This study noted a number of limitations and areas for improvement. First, the assessment only covered 50 HCFs designated as COVID-19 isolation facilities in seven provinces. Accordingly, study findings may not be generalizable. The Vital Medicines Availability and Health Service (VMAHS) survey, a quarterly survey including over 1300 HCFs, may be better suited to review the status of WASH service coverage at the national level and inform policy discussions. Second, this study only applied WASH FIT methodology to assess the status of WASH conditions in HCFs, without implementing other steps such as the establishment of trained WASH FIT teams and the development of improvement plans for each HCF. Consequently, HCF management teams may not be fully owning the WASH FIT methodology for continuous improvements. Third, the assessment did not verify how each HCF managed gaps in WASH services and what guidelines, checklists, and assessment tools were used. Lastly, WASH FIT data were collected between June and September 2020, before COVID-19 response interventions were fully implemented. Thus, the status of WASH service coverage may have improved after the WASH FIT assessment.

## 6. Conclusions

Despite these limitations, this study revealed that WASH FIT is a useful tool to assess the status of WASH services in HCFs in Zimbabwe, even during the COVID-19 pandemic. In Zimbabwe, many HCFs designated as COVID-19 isolation facilities still struggle with access to key WASH services. Immediate WASH interventions are needed to minimize the risk of COVID-19 transmission and other infectious agents within HCFs. By scaling up WASH FIT and mobilizing financial resources, HCFs may be able to monitor WASH service gaps more systematically and address issues in a timely manner.

## Figures and Tables

**Figure 1 ijerph-18-05641-f001:**
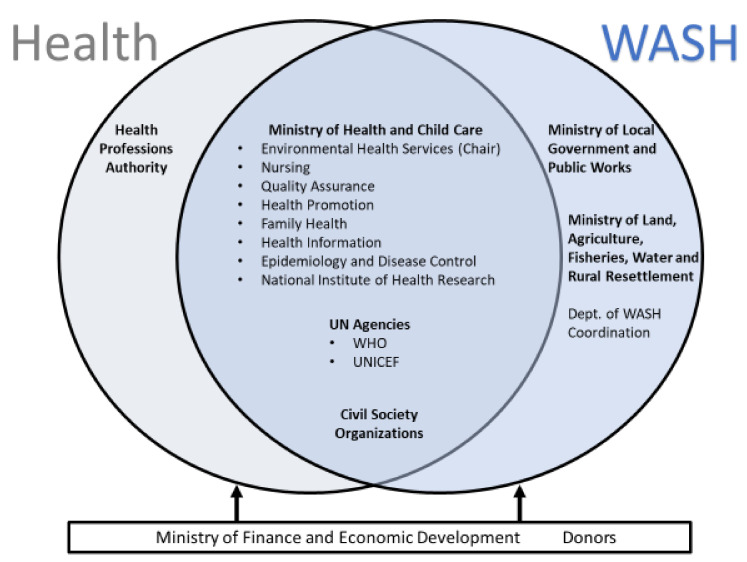
WASH in Healthcare Facilities Taskforce Members in Zimbabwe.

**Figure 2 ijerph-18-05641-f002:**
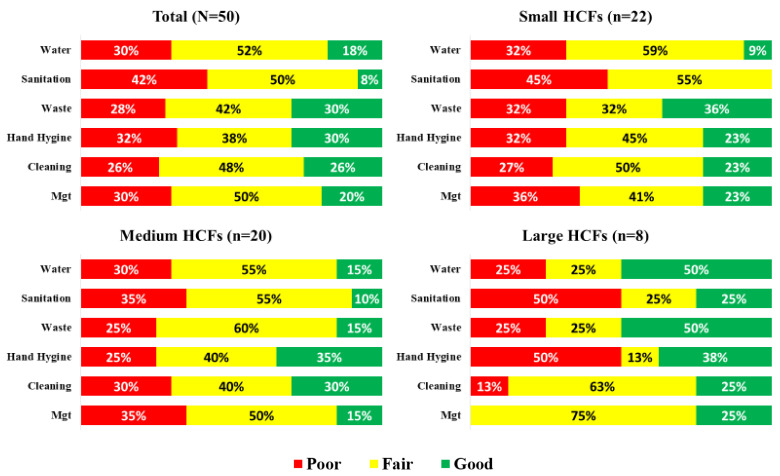
Proportion of HCFs with poor, fair, and good WASH service coverage by the size of HCFs.

**Table 1 ijerph-18-05641-t001:** Key characteristics of 50 health care facilities.

Variables	n (%)
**Province**	
Bulawayo	3 (6)
Manicaland	5 (10)
Mashonaland Central	5 (10)
Mashonaland East	3 (6)
Masvingo	9 (18)
Matabeleland North	8 (16)
Matabeleland South	7 (14)
Midlands	10 (20)
**Hospital size**	
Small (<100 beds)	22 (44)
Medium (100–199 beds)	20 (40)
Large (200 beds or more)	8 (16)
**HCF type**	
Public hospital	40 (80)
Private hospital	3 (6)
Public health center/post	5 (10)
Private health center/post	2 (4)
**Type of Water Source**	
Public taps/standpipe	23 (46)
Hand pumps/boreholes	14 (28)
Piped connection	11 (22)
Surface water	1 (2)
Other	1 (2)
**Type of Sanitation Facility**	
Flush toilet	43 (86)
Pit latrine with a slab	5 (10)
Ventilated improved pit	1 (2)
Pit latrine without a slab	1 (2)
	**mean (std)**
**Daily inpatients**	37.0 (51.3)
**Daily outpatients**	67.9 (67.9)
**Number of beds**	138.7 (164.2)

**Table 2 ijerph-18-05641-t002:** WASH FIT assessment results of 50 health care facilities on essential indicators.

WASHFIT Essential Indicators	Total	Small	Medium	Large	Ratings
**Water**					
Improved water supply piped into the facility or on premises and available	1.44	1.23	1.60	1.63	Fair
Water services available at all times and of sufficient quantity for all uses	0.94	0.91	0.85	1.25	Poor
A reliable drinking water station is present and accessible for staff, patients and care takers at all times and in all locations/wards	1.12	1.00	1.05	1.63	Fair
Drinking water is safely stored in a clean bucket/tank with cover and tap	1.36	1.23	1.35	1.75	Fair
**Sanitation**					
Number of available and usable toilets or improved latrines for patients	1.04	1.09	0.95	1.13	Fair
Toilets or improved latrines clearly separated for staff and patients	1.16	1.23	1.05	1.25	Fair
Toilets or improved latrines clearly separated for male and female	1.52	1.45	1.70	1.25	Good
At least one toilet or improved latrine provides the means to manage menstrual hygiene needs	0.56	0.68	0.45	0.50	Poor
At least one toilet meets the needs of people with reduced mobility	0.34	0.18	0.50	0.38	Poor
Functioning hand hygiene stations within 5 meters of latrines	1.14	1.09	1.10	1.38	Fair
**Health Care Waste**					
A trained person is responsible for the management of health care waste in the health care facility	1.36	1.55	1.20	1.25	Fair
Functional waste collection containers in close proximity to all waste generation points for: non-infectious (general) waste, infectious waste, and sharps waste	0.98	0.77	1.05	1.38	Poor
Waste correctly segregated at all waste generation points	1.04	0.95	1.05	1.25	Fair
Functional burial pit/fenced waste dump or municipal pick-up available for disposal of non-infectious waste	1.46	1.45	1.35	1.75	Fair
Incinerator or alternative treatment technology for the treatment of infectious and sharp waste is functional and of a sufficient capacity	1.26	1.32	1.15	1.38	Fair
Sufficient energy available for incineration or alternative treatment technologies	0.88	0.76	0.72	0.99	Poor
**Hand Hygiene**					
Functioning hand hygiene stations are available at all points of care	1.26	1.09	1.40	1.38	Fair
Hand hygiene promotion materials clearly visible and understandable at key places	1.22	1.09	1.40	1.13	Fair
**Facility Environment, Cleanliness and Disinfection**					
The exterior of the facility is well-fenced, kept generally clean (free from solid waste, stagnant water, no animal and human feces in or around the facility premises)	1.50	1.50	1.45	1.63	Good
General lighting sufficiently powered and adequate to ensure safe provision of health care including at night	1.20	1.23	1.15	1.25	Fair
Floors and horizontal work surfaces appear clean	1.42	1.23	1.70	1.25	Fair
Appropriate and well-maintained materials for cleaning are available	1.06	0.86	1.15	1.38	Fair
At least two pairs of household cleaning gloves and one pair of overalls or apron and boots in a good state for each cleaning and waste disposal staff member	0.88	0.77	1.05	0.75	Poor
At least one member of staff can demonstrate the correct procedures for cleaning and disinfection and apply them as required to maintain clean and safe rooms	1.46	1.45	1.50	1.38	Fair
Beds have insecticide treated nets to protect patients from mosquito-borne diseases	0.72	0.50	1.00	0.63	Poor
**Management**					
WASH FIT or other quality improvement/management plan for the facility is in place, implemented and regularly monitored	0.94	0.86	1.05	0.88	Poor
An annual planned budget for the facility is available and includes funding for WASH infrastructure, services, personnel and the continuous procurement of WASH items, which is sufficient to meet the needs of the facility	0.72	0.50	0.75	1.25	Poor
An up-to-date diagram of the facility management structure is clearly visible and legible	1.30	1.05	1.45	1.63	Fair
Adequate cleaners and WASH maintenance staff are available	1.14	1.14	1.15	1.13	Fair

## Data Availability

Data is presented in the article and Appendix A.

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
