# Peer review of "Using the Water and Sanitation for Health Facility Improvement Tool (WASH FIT) in Zimbabwe: A Cross-Sectional Study of Water, Sanitation and Hygiene Services in 50 COVID-19 Isolation Facilities"

_ijerph, 2021, doi:10.3390/ijerph18115641_

Round 1

Reviewer 1 Report

Safe water, sanitation and hygiene services including  good hand hygiene are extremely important when working with COVID-19 patients in order to protect health care personal and other patients and other personal.

Title: A good title is short. Would a better title be:  Water, Sanitation and Hygiene Services for Health Facility Improvement Tool (WASH FIT) in 50 COVID-19 isolation facilities in Zimbabwe

Abstract includes all necessary things and it is painful to read that some of these facilities are not available in all hygiene facilities. It is also often found that toilets do not well serve people with reduced mobility and women with menstrual periods, but it is good to mention this again.

Material and methods: The most important things are described. You could describe more your evaluation forms since people from industrial countries who have never visited in developing countries have difficulty to image the problems when there are no necessary facilities.  What kind of claims there was needed to evaluate some activity as 2?

How long time the patients of COVID-19 spent in the isolation facilities? How high percentage of patients were infected by corona virus with symptoms?  

In some cases (as line 78) your text is rather complicated.

It is usual that water coverage is highly prioritized so that water is available in 81 % of cases (as you refer) but sanitation in only 17 % of cases and hygiene services in 58 % of cases. Sanitation is often the weakest point. Hygiene services may be partly newer and there are many new things to that it may be difficult to follow all novel things – as now hand disinfection compounds in shops and homes (in my country). What is energye (line 189)?

Figure 2 is beautiful. Anyhow, it is important to notice that the number of large hospitals is only eight. Thus only one hospital means 13  % and two means 25 %. You could have merged medium and large-size hospitals.

I agree that the results and findings serve mainly this area and this time. Anyhow, this work can serve many other areas and give ideas about what could be done in the other areas. Hopefully, already these results in a possible more serious COVID-19 situation in 2021 may increase sanitation and hygiene services and keep water safe.  Sometimes when hospitals or different organs can compare their own results to the results of their neighbors, they can see what they should improve without any costs or with very low costs. Thus for instance it would be very easy and cheap to serve better menstruating women and girls with a simple waste bin and water for washing and a small shelf to store the clean napkin.  The bin should have a lid but both the bin and its lid can be old. If the wasted napkins will be left in a bin there will no more be risks caused by wasted napkins.   Open discussion may also help that all personal will understand the importance of their own work.   

Reviewer 2 Report

Title: Using the Water and Sanitation for Health Facility Improvement Tool (WASH FIT) in Zimbabwe: A cross-sectional study 2 of water, sanitation and hygiene services in 50 COVID-19 isolation facilities.

This study aimed to describe the status of WASH services in 50 COVID-19 isolation facilities and discuss how WASH in HCF Taskforce members may contribute to addressing key gaps identified.

The study methodology relied on using quantitative data generated through the use of the WASH FIT multi-step framework to assess 50 Healthcare Facilities to collect key information on the status of WASH services.

The presentation of the results was good and there was a clear link between the introduction, results and discussion of findings. The discussion of the results is excellent, with relevant issues raised.

Overall the paper presented is well written, but in the literature review the authors mention that “What literature has not elucidated is if the existing WASH assessment methodology can be useful under the COVID-19 response and how countries may accelerate WASH interventions in HCFs without taking a siloed approach by health and WASH sectors” however there is a recent study which has done this, and more comparison to it could be made in the literature review and discussion;

Ashinyo, M. E., Amegah, K. E., Dubik, S. D., Ntow-Kummi, G., Adjei, M. K., Amponsah, J., ... & Akoriyea, S. K. Research Paper Evaluation of water, sanitation and hygiene status of COVID-19 healthcare facilities in Ghana using the WASH FIT approach.
